# Information Theory Meets Quantum Chemistry: A Review and Perspective

**DOI:** 10.3390/e27060644

**Published:** 2025-06-16

**Authors:** Yilin Zhao, Dongbo Zhao, Chunying Rong, Shubin Liu, Paul W. Ayers

**Affiliations:** 1Department of Chemistry and Chemical Biology, McMaster University, Hamilton, ON L8S 4M1, Canada; zhaoyilin10@foxmail.com; 2Institute of Biomedical Research, Yunnan University, Kunming 650500, China; dongbo@ynu.edu.cn; 3College of Chemistry and Chemical Engineering, Hunan Normal University, Changsha 410081, China; rongchunying@aliyun.com; 4Department of Chemistry, University of North Carolina, Chapel Hill, NC 27599-3420, USA; 5Research Computing Center, University of North Carolina, Chapel Hill, NC 27599-3420, USA

**Keywords:** quantum chemistry, information theory, classical information theory, quantum information theory, Shannon entropy, von Neumann entropy, relative entropy, joint entropy, conditional entropy, mutual information, information-theoretic approach, orbital entanglement

## Abstract

In this survey, we begin with a concise introduction to information theory within Shannon’s framework, focusing on the key concept of Shannon entropy and its related quantities: relative entropy, joint entropy, conditional entropy, and mutual information. We then demonstrate how to apply these information-theoretic tools in quantum chemistry, adopting either classical or quantum formalisms based on the choice of information carrier involved.

## 1. Motivation

Information theory was first established in the 1920s through the works of Harry Nyquist and Ralph Hartley [1,2] and propelled to prominence in the 1940s by Claude Shannon [3]. The concept of information theory, which encompasses the quantification, storage, and communication of information [4,5], is too broad to be simply described. Today, information theory serves as a versatile tool in statistics, natural sciences, machine learning, quantum computing, and numerous other fields.

In molecular electronic structure theory, we routinely encounter sets of non-negative values that sum to unity, corresponding to valid probability distributions. This includes properly normalized electron density distributions, the eigenvalues of the reduced density matrix, the squared modulus of wave function coefficients in an orthonormal basis, and more [6,7]. Information theory can be used to analyze these probability distributions, an approach that has been actively pursued by researchers with remarkable success since the 1970s [8,9,10,11,12,13,14,15,16,17,18,19,20,21,22,23,24,25,26,27,28,29,30,31,32,33,34]. Incorporating concepts from information theory into quantum chemistry has provided valuable insights into the nature and behavior of electronic systems.

The two most popular approaches to electronic structure theory are quantum many-body theory [35,36,37,38,39,40,41,42,43,44,45,46,47,48,49,50] and density functional theory (DFT) [51,52,53,54]. Integrating these strategies with classical information theory (CIT) leads to what is normally called the information-theoretic approach (ITA) [8,9,10,11,12,13,14,25,26,27,28,29,30,31,32,33,34]; integrating these strategies with quantum information theory (QIT) [15,16,17,18,19,20,21,22,23,24] leads to a different set of tools, with different interpretations and utility. Some key concepts of classical and quantum information theory in quantum chemistry that we will discuss in this article are summarized in Figure 1. Although information theory extends far beyond Shannon’s work, other frameworks such as Rényi and Fisher information have their own application domains [55,56,57,58]. Due to space limitations, our discussion in this paper will be confined to Shannon’s framework, although we note that the Rényi formulation and Tsallis formulations [55,57], in particular, reduce to Shannon’s in appropriate limits and, as such, the following analysis can be generalized to these quantities.

This review concisely introduces some fundamental aspects of information theory and its applications in quantum chemistry. Section 2 discusses the basic concepts of Shannon entropy and its related quantities: relative entropy, joint entropy, conditional entropy, and mutual information. To integrate quantum information theory into quantum chemistry, we introduce the reduced density matrix (RDM) in Section 3, which serves as a powerful tool to simplify the complexity of quantum states while retaining essential information about the state of a subsystem. Section 4 introduces the application of information theory in quantum chemistry; for the classical approach, we convert the reduced density matrix in the position representation to derive the electron density and pair density, which act as information carriers for classical information theory. We also explore the corresponding quantum concepts in Section 5, such as von Neumann entropy and quantum mutual information, which are used to analyze the entanglement in quantum many-body systems.

## 2. Brief Introduction to Information Theory

### 2.1. Shannon Entropy

One of the core concepts in information theory is the Shannon entropy, named after Claude Shannon, the founder of information theory [3]. Let X be a discrete random variable with alphabet X and probability distribution function p(x)=Pr{X=x} for x∈X. The Shannon entropy is defined as:(1)$$\begin{array}{cc} \;H_{\left(X\right)} & =-\sum_{x\in X}p\left(x\right)\log p\left(x\right)\\ \; & =E_{p}\left[\log\frac{1}{p(X)}\right] \end{array}$$
where the expectation is denoted by E. Thus, if X∼p(x), the expected value of the random variable g(X) is written as:(2)$$E_{p}\left[g(X)\right]=\sum_{x\in X}g\left(x\right)p\left(x\right)$$

The Shannon entropy admits multiple interpretations, one of which states that it provides a mathematical framework to measure the uncertainty of a random event. The term *entropy* originates from the Greek word *trope* (meaning change) and was first introduced by Clausius in 1854 in the context of the second law of thermodynamics [59]. Shannon himself explained his rationale for adopting this term in a disarming way:

My greatest concern was what to call it. I thought of calling it ‘information’, but the word was overly used, so I decided to call it ‘uncertainty’. When I discussed it with John von Neumann, he had a better idea. Von Neumann told me, “You should call it entropy, for two reasons. In the first place your uncertainty function has been used in statistical mechanics under that name, so it already has a name. In the second place, and more important, nobody knows what entropy really is, so in a debate you will always have an advantage [60].

### 2.2. Relative Entropy

To quantify how distinguishable two probability distributions are, by introducing a second probability distribution q(x)=Pr{X=x}, where x∈X, we now look for a distinguishability measure of those two probability distributions [61].

Analogous to the concept of *distance* (norm) in Euclidean space, information theory defines the relative entropy (information divergence) $D_{\left(P||Q\right)}$ between two probability distributions p(x) and q(x). Among the most popular divergence measures include the Bregman divergences [62,63] and *f*-divergences [64,65,66]. The *f*-divergence is defined as follows:(3)$$\;D_{f(P||Q)}=\sum_{x\in X}f\left(\frac{p(x)}{q(x)}\right)q\left(x\right)$$
where f(x) is a convex function satisfying f(1)=0. The Bregman divergence is defined as follows:(4)$$\;D_{F(P||Q)}=F\left(P\right)-F\left(Q\right)-\sum_{x\in X}\frac{\delta F[p(x)]}{\delta q(x)}\left(p(x)-q(x)\right)$$
where F[p] is a convex functional.

Most measures of relative entropy fit these frameworks, but the Kullback–Leibler (KL) divergence [67] is both an *f*-divergence and a Bregman divergence. Therefore, the Kullback–Leibler divergence is the most widely used definition of relative entropy, with its expression given by the following: (5)$$\begin{array}{cc} \;D_{KL(P||Q)} & =\sum_{x\in X}p\left(x\right)\log\left(\frac{\begin{array}{c} p(x) \end{array}}{\begin{array}{c} q(x) \end{array}}\right)\\ \; & =E_{p}\left[\log\frac{p_{\left(X\right)}}{q_{\left(X\right)}}\right] \end{array}$$

Formally, a metric (such as the Euclidean norm) must satisfy four axioms for all distributions P, Q, and R: [68]

Non-negativity: d(P,Q)≥0Identity of indiscernibles: d(P,Q)=0⇔P=Q.Symmetry: d(P,Q)=d(Q,P)Triangle inequality: d(P,Q)≤d(P,R)+d(R,Q)

Figure 2 provides an intuitive illustration of the non-symmetric nature of the Kullback–Leibler divergence. Additionally, counterexamples exist where(6)$$D_{KL(P||Q)}>D_{KL(P||R)}+D_{KL(R||Q)}$$
demonstrating that the Kullback–Leibler divergence violates the triangle inequality. Thus, the Kullback–Leibler divergence is considered a premetric but not a full metric.

### 2.3. Bivariate Entropy

To study correlations between variables, it is useful to extend the Shannon entropy to the multivariate case. For simplicity, consider a pair of variables (X,Y). Then, bivariate information-theoretic quantities such as joint entropy, conditional entropy, and mutual information can be defined, and their relationships are illustrated in Figure 3.

The most straightforward two-variable extension of Shannon entropy is the joint entropy $H_{\left(X,Y\right)}$, which measures the total uncertainty when considering a pair of variables together, and its corresponding expression is as follows:(7)$$\begin{array}{cc} \;H_{\left(X,Y\right)} & =-\sum_{x\in X}\sum_{y\in Y}p\left(x,y\right)\log p\left(x,y\right)\\ \; & =E_{p}\left[\log\frac{1}{p_{\left(X,Y\right)}}\right] \end{array}$$
where p(x,y) is the joint probability distribution function. Since a pair of variables can be treated as a single vector of length two, this extension does not introduce fundamentally new concepts.

Other bivariate entropy measures include conditional entropy and mutual information, which can be expressed in the framework of the Kullback–Leibler divergence we introduced in Equation (Equation 5). The conditional entropy of one variable, given another, is defined as the expected value of the entropy of the conditional distributions, averaged over the conditioning variable.(8)$$\begin{array}{cc} \;H_{\left(Y|X\right)} & =\sum_{x\in X}p\left(x\right)H_{\left(Y|X=x\right)}\\ \; & =-\sum_{x\in X}p\left(x\right)\sum_{y\in Y}p\left(y|x\right)\log p\left(y|x\right)\\ \; & =-\sum_{x\in X}\sum_{y\in Y}p\left(x,y\right)\log p\left(y|x\right)\\ \; & =E_{p}\left[\log\frac{1}{p_{\left(Y|X\right)}}\right] \end{array}$$ Compared to the concept of entropy, which is a probabilistic measure of uncertainty; information is a measure of a reduction in that uncertainty. The mutual information is a measure of the amount of information that one variable contains about another. It is defined as follows:(9)$$\begin{array}{cc} \;I_{\left(X;Y\right)} & =D_{KL(p(x,y)||p(x)p(y))}\\ \; & =\sum_{x\in X}\sum_{y\in Y}p\left(x,y\right)\log\frac{p(x,y)}{p(x)p(y)}\\ \; & =E_{p}\left[\log\frac{p_{\left(X,Y\right)}}{p_{\left(X\right)}p_{\left(Y\right)}}\right] \end{array}$$

The conditional and mutual information, defined using the Kullback–Leibler divergence, exhibit several important properties, enabling a clear and rigorous analysis.

The chain rule.(10)$$H_{\left(X,Y\right)}=H_{\left(X\right)}+H_{\left(Y|X\right)}$$Subadditivity.(11)$$H_{\left(X,Y\right)}\leq H_{\left(X\right)}+H_{\left(Y\right)}$$The relationship between different bivariate entropy,(12)$$\begin{array}{cc} I_{\left(X;X\right)} & =H_{\left(X\right)} \end{array}$$(13)$$\begin{array}{cc} I_{\left(X;Y\right)} & =I_{\left(Y;X\right)} \end{array}$$(14)$$\begin{array}{cc} I_{\left(X;Y\right)} & =H_{\left(X\right)}-H_{\left(X|Y\right)} \end{array}$$(15)$$\begin{array}{cc} I_{\left(X;Y\right)} & =H_{\left(Y\right)}-H_{\left(Y|X\right)} \end{array}$$(16)$$\begin{array}{cc} I_{\left(X;Y\right)} & =H_{\left(X\right)}+H_{\left(Y\right)}-H_{\left(X,Y\right)} \end{array}$$(17)$$\begin{array}{cc} I_{\left(X;Y\right)} & =H_{\left(X,Y\right)}-H_{\left(X|Y\right)}-H_{\left(Y,X\right)} \end{array}$$

## 3. Basic Ingredients of Information Theory in Quantum Chemistry: Reduced Density Matrix

First proposed by Paul Dirac in 1930 [69], the reduced density matrix (RDM) becomes a fundamental tool in quantum chemistry and many-body physics, enabling the analysis of subsystems within larger quantum systems [70,71,72,73,74]. Because electron–electron repulsion is a two-body operator, the one- and two-electron reduced density matrices (1-RDM and 2-RDM) are the quantities of greatest interest in quantum chemistry, as it suffices to determine most molecular properties, including the electronic energy [75,76,77]. Similarly, for the application of information theory in quantum chemistry, the reduced density matrix emerges as the natural mathematical framework for elucidating the correlations and entanglement [78,79].

### 3.1. Density Matrix and Reduced Density Matrix

For quantum many-body systems, other than the usual state vectors |Ψ〉 (mathematically described as rays in a projective Hilbert space), quantum states can also be represented by the density matrix D (density operator $\hat{D}$), which unifies the description of pure and mixed states. If a density matrix is obtained as(18)D=|Ψ〉〈Ψ|
then the quantum state is a pure state. A mixed state cannot be expressed this way and instead requires a statistical mixture,(19)$$\;D=\sum_{n}\omega_{n}|\Psi_{n}\rangle \langle \Psi_{n}|\;\left(\omega_{n}\geq 0,\;\sum_{n}\omega_{n}=1\right)$$
where $\left\{|\Psi_{n}\rangle \right\}$ is an ensemble of pure states with weights (probability) $\omega_{n}\geq 0$.

To further elucidate the concepts of pure states and mixed states, we employ a two-state quantum system as an illustrative example. It is often implemented using a spin $\frac{1}{2}$ physical particle, with two basis states {|0〉,|1〉}. A pure state of such a two-state quantum system is represented by a state vector:(20)$$\;|\psi \rangle =\alpha |0\rangle +{\beta |1\rangle \;|\alpha |}^{2}+{\left|\beta \right|}^{2}=1\;\alpha ,\beta \in C$$ The constraint conditions in Equation (Equation 20) indicate that a pure state of a two-state quantum system can be equivalently expressed as(21)$$\;|\psi \rangle =\cos\left(\frac{\theta }{2}\right)|0\rangle +e^{i\varphi }\sin\left(\frac{\theta }{2}\right)|1\rangle \;\theta \in \left[0,\pi \right],\varphi \in \left[0,2\pi \right]$$ Thus, the pure state of a two-state quantum system can be uniquely determined by two parameters θ and φ. Here, we can establish a geometric representation for this two-state quantum system: every pure state |ψ〉 defined in Equation (Equation 21) uniquely corresponds to a point (cosφsinθ, sinφsinθ, cosθ) on the surface of the Bloch sphere as shown in Figure 4.

For the mixed states (the same applies to pure states) of a two-state quantum system, it is evident that the density matrix can be represented by a 2 × 2 matrix, since(22)$$\begin{array}{cc} D & =\sum_{n}\omega_{n}|\psi_{n}\rangle \langle \psi_{n}|\\ \; & =\sum_{n}\omega_{n}\left(\right|\alpha_{n}{|}^{2}\left|0\rangle \langle 0\right|+\alpha_{n}\beta_{n}^{*}\left|0\rangle \langle 1\right|+\alpha_{n}^{*}\beta_{n}\left|1\rangle \langle 0|+\right|\beta_{n}{|}^{2}|1\rangle \langle 1|)\\ \; & =\left[\begin{array}{cc} \sum_{n}\omega_{n}{\left|\alpha_{n}\right|}^{2} & \sum_{n}\omega_{n}\alpha_{n}\beta_{n}^{*}\\ \sum_{n}\omega_{n}\alpha_{n}^{*}\beta_{n} & \sum_{n}\omega_{n}{\left|\beta_{n}\right|}^{2} \end{array}\right] \end{array}$$ It can be rigorously proven that every point within the Bloch sphere corresponds to a mixed state. In particular, the center point represents the maximally mixed state, as established in quantum information theory [80].

For conciseness, we will henceforth assume that the density matrix (DM) and the reduced density matrix (RDM) correspond to pure states, but most of the analysis extends directly to mixed states. The *k*-electron reduced density matrix (*k*-RDM) is defined as the partial trace, over N−k dimensions, of an *N*-electron density matrix; it is usually expanded on the basis of single-particle states (i.e., spin orbitals). Specifically,(23)D¯k=∑p1,…,pk,q1,…,qk|p1,…,pk〉〈q1,…,pk|(D¯q1,…,qkp1,…,pkk)
where(24)D¯q1,…,qkp1,…,pkk=〈Ψ|ap1†…,apk†aqk,…,aq1|Ψ〉
are elements of the *k*-RDM, which is Hermitian and positive semidefinite because the density matrix is Hermitian and positive semidefinite. We use the overbar above symbols when we need to explicitly distinguish quantities expressed in the spin-orbital, rather than the spatial orbital, basis.

### 3.2. 1-RDM and 2-RDM

The quantities of greatest interest to chemists are the one- and two-electron reduced density matrices. The 1-RDM is defined as follows:(25)D¯1=∑pq|p〉〈q|〈Ψ|ap†aq|Ψ〉=∑pqD¯qp1|p〉〈q|
and the 2-RDM is defined as follows:(26)D¯2=∑pqrs|pq〉〈rs|〈Ψ|ap†aq†asar|Ψ〉=∑pqrsD¯rspq2|pq〉〈rs|

In our convention, the trace of the 1-RDM is the number of electrons and the trace of the 2-RDM is the number of electron pairs:(27)Tr[D¯1]=N(28)Tr[D¯2]=N2=N(N−1)2 The reader is cautioned that some authors use different normalization conventions (e.g., unit normalization or the number of non-unique electron pairs, Tr[D¯2]=N(N−1)).

The reduced density matrix (RDM) can be represented in a spin block format because the projection of the spin vector onto a specified axis, ${\hat{S}}^{z}$, commutes with the molecular Hamiltonian.(29)$$\left[\hat{H},{\hat{S}}^{z}\right]=\hat{H}{\hat{S}}^{z}-{\hat{S}}^{z}\hat{H}=0$$ Thus, the one-particle reduced density matrix (1-RDM) will have a block-diagonal form. (30)D¯1=D¯αα100D¯ββ1 Similarly, the spin-block format of the two-particle reduced density matrix (2-RDM) is as follows: (31)D¯2=D¯αααα20000D¯αβαβ2D¯βααβ200D¯αββα2D¯βαβα20000D¯ββββ2 Note that knowledge of any one of the four opposite-spin blocks, e.g., D¯βααβ2, suffices to determine the others using (anti)symmetry.

### 3.3. 3-RDM and 4-RDM

In general, reduced density matrices of order greater than two are essentially redundant for most quantum chemistry problems, as electrons interact only pairwise. However, for certain niche quantum chemistry applications, the information provided by the 3-RDM and 4-RDM is still required. The three-electron reduced density matrix (3-RDM) can be explicitly defined as follows:(32)D¯3=∑pqrstu|pqr〉〈stu|〈Ψ|ap†aq†ar†auatas|Ψ〉=∑pqrstuD¯stupqr3|pqr〉〈stu|
and, similarly, the four-electron reduced density matrix (4-RDM) is defined as follows:(33)D¯4=∑pqrstuvw|pqrs〉〈tuvw|〈Ψ|ap†aq†ar†as†awavauat|Ψ〉=∑pqrstuvwD¯tuvwpqrs4|pqrs〉〈tuvw|

We can easily determine lower-order reduced density matrices (RDM) by taking the partial trace of a higher-order one. However, using higher-order reduced density matrices than required is undesirable, as the computational cost and memory requirements of high-order reduced density matrices are prohibitively large. For example, the storage required to store the complete 3-RDM and 4-RDM scales as $O(n^{6})$ and $O(n^{8})$, respectively, where *n* is the number of basis functions. Higher-order reduced density matrices can be systematically, but approximately, expressed in terms of lower-order reduced density matrices using diagrammatic and statistical techniques [81,82,83,84,85,86,87,88,89,90,91].

Specifically, using the cumulant expansion one can decompose higher-order RDM into sums of products of lower-order quantities and nonreducible k-order cumulants Δk [81,82,88,92]. As a starting point, the 1-RDM can be expressed as the sum of a mean-field term and a correlated cumulant term:(34)Dqp1=(D01)qp+Δqp1 where D01 is the known mean-field 1-RDM, the two-particle reduced density matrix (2-RDM) can then be expressed as the wedge (or Grassmann) product of two one-particle reduced density matrices (1-RDM) and a cumulant 2-RDM.(35)Drspq2=2Drp1∧Dsq1+Δrspq2
where the wedge product, denoted as ∧, is defined as an antisymmetric tensor product. Given two reduced density matrices of orders k and l, denoted Dk and Dl respectively, their wedge product yields a (k + l)-order RDM satisfying (36)Dk∧Dl=1(k+l)!∑πsgn(π)Pπ(Dk⊗Dl) where $P_{\pi }$ permutes the upper and lower indices according to the permutation π. sgn(π) is the parity of the permutation (+1 for even, −1 for odd). As the most elementary example, the wedge product of two 1-RDMs is given by
(37)Drp1∧Dsq1=Drp1Dsq1−Dsp1Drq1 The cumulant expansions of the three-particle and four-particle reduced density matrices (3-RDM and 4-RDM) are given by(38)Dstupqr3=6Dsp1∧Dtq1∧Dur1+9Dstpq2∧Dur1+Δstupqr3
and(39)Dtuvwpqrs4=24Dtp1∧Duq1∧Dvr1∧Dws1+72Dtupq2∧Dvr1∧Dws1+24Dtupq2∧Dvwrs2+16Dtuvpqr3∧Dws1+Δtuvwpqrs4 The traces of the density matrices in these expressions give the number of distinguishable pairs, triples, and quartets of electrons, respectively. Dividing by k!, where *k* is the order of the reduced density matrix, recovers the definition we’ve used elsewhere.

## 4. Classical Information Theory in Quantum Chemistry

The Hohenberg–Kohn theorems [51,52,93,94,95] imply that all ground state properties are functionals of the electron density; this establishes the theoretical basis for using classical information theory that extracts chemical insights by treating electron densities as probability distributions.

### 4.1. Electron Density in Position Space

In Section 3, we establish reduced density matrices as the basic ingredients for information-theoretic analysis in quantum chemistry. For molecular systems, the concept of probability distribution p(x) in general information theory is specialized as an analytical functional of the electron density ρ(r), which depends solely on the spatial coordinates, $r\in R^{3}$:(40)$$\;\rho \left(r\right)=N\int |\Psi \left(x_{1},x_{2},\ldots ,x_{N}\right){|}^{2}ds_{1}dx_{2}\ldots dx_{N}$$ By construction,(41)ρ(r)≥0,∫ρ(r)dr=N Another fundamental information carrier in quantum chemistry is the two-electron distribution function, or pair density, defined as(42)$$\;\Gamma \left(r_{1},r_{2}\right)=\frac{N(N-1)}{2}\int {\left|\Psi \left(x_{1},x_{2},\ldots ,x_{N}\right)\right|}^{2}ds_{1}ds_{2}dx_{3}\ldots dx_{N}$$ By definition,(43)$$\Gamma \left(r_{1},r_{2}\right)\geq 0,\;\;\int \int \;\Gamma \left(r_{1},r_{2}\right)dr_{1}dr_{2}=\frac{N(N-1)}{2}$$ Analogous to density-functional theory, a complete description of electronic structure can be constructed based on the pair density (and also higher-order electron distribution functions) using appropriate generalizations of the Hohenberg–Kohn theorem [96,97,98,99,100,101,102,103,104].

The electron density and pair density can be computed from the reduced density matrices (RDMs) we introduced in Section 3. However, in that section, we considered the spin-resolved reduced density matrices, and it is more convenient in this context to trace out the spin coordinates and obtain a representation of the reduced density matrices in terms of spatial orbitals, i.e., (44)D1=D¯αα1+D¯ββ1(45)D2=D¯αααα2+D¯αβαβ2+D¯βαβα2+D¯ββββ2 Next, we transform the one- and two-electron reduced density matrices from the (second-quantized) spatial-orbital representation into the (first-quantized) position representation, (46)ρ(r;r′)=Dνμ1ϕμ(r)ϕν(r′)(47)Γ(r1,r2;r1′,r2′)=Dκλμν2ϕμ(r1)ϕν(r2)ϕκ(r1′)ϕλ(r2′) We use the indices μ, ν, κ, and λ to label atomic orbitals. The one-electron density ρ(r) (Equation (40)) and the pair-electron density $\Gamma (r_{1},r_{2})$ (Equation (42)) represent the diagonal components of the spinless one- and pair-electron reduced density matrix in position space, respectively, which is obtained by setting the un-primed spatial variables equal to the primed spatial variables, $r_{i}=r_{i}^{\prime }$. Note that off-diagonal elements of the orbital representation of the RDM contribute to diagonal elements of the spatial representation of the RDM and vice versa. This has significant implications for the *N*-representable of electron distribution functions [96,105].

### 4.2. Information-Theoretic Approach Chemical Descriptors

Taking electron density as the information carrier, classical information theory has been applied to DFT since the mid-20th century to study atoms and molecular systems within the framework of the information-theoretic approach (ITA) [8,9,10,11,12,13,14,25,26,27,28,29,30,31,32,33,34].

The Shannon entropy, with ρ(r) as its information carrier, measures the spatial delocalization of the electron density and is defined as follows:(48)$$\;S_{S}\equiv S_{S(X)}=-\int \rho \left(r\right)\log\rho \left(r\right)dr$$ In accordance with the traditions of the information-theoretic approach, the base-10 logarithm is used throughout Section 4.2. In the information-theoretic approach, we employ a broader set of formulas beyond Shannon entropy. To maintain notational clarity, we adopt a systematic symbolic convention where Shannon’s formula is specifically denoted by the symbol $S_{S}$. When only the one-electron density is considered, the subscript_(**X**)_ in Equation (48) is typically omitted.

By introducing the reference density $\rho^{0}\left(r\right)$, which corresponds to the electron density of the promolecule constructed under the assumption that each atom retains its density as if it were isolated [106,107,108,109,110,111], we can define the relative Shannon entropy $S_{S}^{r}$. This quantity, also referred to as the Kullback–Leibler divergence, information gain, or information divergence [14,26,27,111,112,113,114,115], is defined by(49)$$S_{S}^{r}\equiv D_{KL(X||X^{0})}=\int \rho \left(r\right)\log\frac{\rho (r)}{\rho^{0}\left(r\right)}dr$$

The joint entropy $S_{S(X,Y)}$, which measures the localization of the pair of electrons in their respective spaces, is defined as follows:(50)$$S_{S(X,Y)}=-\;\int \int \;\Gamma \left(r_{1},r_{2}\right)\log\Gamma \left(r_{1},r_{2}\right)dr_{1}dr_{2}$$

In addition to joint entropy, other bivariate entropy measures need to maintain the same normalization condition for the state density and the reference state density. Thus, the normalized one- and pair-electron densities $\rho^{\sigma }\left(r\right)$ and $\Gamma^{\sigma }\left(r_{1},r_{2}\right)$ are defined as(51)$$\rho^{\sigma }\left(r\right)\equiv \sigma \left(r\right)=\rho \left(r\right)/\;N$$
and(52)Γσ(r1,r2)=Γ(r1,r2)/N2
where N is the number of electrons. Following the definition of Parr and Bartolotti in 1983 [116], these unit-normalized densities are also referred to as shape functions σ(r) [116,117,118,119]. They exhibit the obvious non-negative properties $\rho^{\sigma }\left(r\right)\geq 0\;\forall r$ and $\Gamma^{\sigma }\left(r_{1},r_{2}\right)\geq 0\;\forall \left\{r_{1},r_{2}\right\}$.

Using the normalized pair electron density as the distribution functions, the joint entropy $S_{S(X,Y)}^{\sigma }$ is defined as(53)$$S_{S(X,Y)}^{\sigma }=-\;\int \int \;\Gamma^{\sigma }\left(r_{1},r_{2}\right)\log\Gamma^{\sigma }\left(r_{1},r_{2}\right)dr_{1}dr_{2}$$ The conditional entropy is defined as the Kullback–Leibler divergence of the unit-normalized pair electron density from the unit-normalized electron densities.(54)$$\begin{array}{cc} S_{S(X|Y)}^{\sigma } & =-D_{KL(\Gamma^{\sigma }\left(r_{1},r_{2}\right)||\rho^{\sigma }\left(r_{2}\right))}\\ \; & =-\int \rho^{\sigma }\left(r_{2}\right)dr_{2}\int \Gamma^{\sigma }\left(r_{1}|r_{2}\right)\log\Gamma^{\sigma }\left(r_{1}|r_{2}\right)dr_{1}\\ \; & =-\;\int \int \;\Gamma^{\sigma }\left(r_{1},r_{2}\right)\log\frac{\Gamma^{\sigma }\left(r_{1},r_{2}\right)}{\rho^{\sigma }\left(r_{2}\right)}dr_{1}dr_{2} \end{array}$$ The mutual information $I_{\left(X;Y\right)}$, is defined as the Kullback–Leibler divergence of the unit-normalized pair electron density from the product of two unit-normalized electron densities. The mutual information then measures the divergence of the pair electron density from the value it would have if the electrons moved entirely independently.(55)$$\begin{array}{cc} S_{S(X;Y)}^{\sigma }\equiv I_{\left(X;Y\right)} & =D_{KL(\Gamma^{\sigma }\left(r_{1},r_{2}\right)||\rho^{\sigma }\left(r_{1}\right)\rho^{\sigma }\left(r_{2}\right))}\\ \; & =\;\int \int \;\Gamma^{\sigma }\left(r_{1},r_{2}\right)\log\frac{\Gamma^{\sigma }\left(r_{1},r_{2}\right)}{\rho^{\sigma }\left(r_{1}\right)\rho^{\sigma }\left(r_{2}\right)}dr_{1}dr_{2} \end{array}$$
Within the information theory framework introduced in Section 2, which utilizes the one- and pair-electron densities as fundamental information carriers, all the classical information-theoretic quantities we defined in this subsection preserve the essential mathematical properties we introduced in Section 2.3.

#### Examples and Illustrations

Using the information-theoretic approach, we can systematically interpret chemical concepts such as chemical bonds, chemical reactivity, electron shells, lone electron pairs, and more [8,9,10,12,13,14,120,121,122,123,124,125,126,127,128,129,130]. The information-theoretic approach descriptors can be categorized based on the scope of molecular features they capture:Global Descriptors: Assign a value to the entire system.Local Descriptors: Assign a value to each position in the system.Non-local Descriptors: Assign a value to each pair of positions in the system.

The global ITA descriptors describe the overall properties of the system as a whole, enabling prediction of properties such as polarizability, aromaticity, acidity/basicity, and reactivity [9,10,12,120,121,122,123,131,132]. Figure 5 plots the correlation between Shannon entropy aromaticity ($\Delta S_{S}$) [120] and several established aromaticity indices: the harmonic oscillator model of aromaticity (HOMA) [133,134], the aromatic fluctuation index (FLU) [135], and nucleus-independent chemical shifts (NICS) [136,137]. The strong correlations demonstrate that Shannon entropy effectively characterizes aromaticity.

The local ITA descriptors can serve as regioselectivity indicators, through a coarse-graining process, which involves integrating their values over atoms, functional groups, or fragment regions; condensed descriptors are obtained [131,132,133]. These descriptors help identify the most reactive atoms, functional groups, or bonds. Figure 6 provides an example of how local Shannon entropy is applied to reveal electron shell structures in noble gas atoms [125]. The radial distribution functions of the Shannon entropy densities are defined as(56)$$\begin{array}{cc} {\bar{h}}_{\left(X\right)}\left(r\right) & =\int r^{2}h_{\left(X\right)}\left(r,\theta ,\phi \right)\sin\left(\theta \right)d\theta d\phi \\ \; & =4\pi r^{2}h_{\left(X\right)}\left(r\right) \end{array}$$ It shows step-like increases in entropy at shell boundaries.

Non-local ITA descriptors quantify how the properties of a molecule at one location respond to changes at another distant point within the same molecule. These descriptors can also be condensed into response matrices, to quantify the correlation between fragments of the system [139]. Figure 7 illustrates the application of joint entropy, conditional entropy, and the mutual information kernel to analyze electron correlation in the krypton atom.

## 5. Quantum Information Theory in Quantum Chemistry

In this section, we transition our discussion of information theory to the quantum realm [21,23,24,47,78,79,80,140,141,142,143,144,145,146,147,148,149,150,151,152,153,154,155]. As we delve deeper, it will become evident that the quantum case holds far richer possibilities, primarily due to the superposition principle.

### 5.1. Bipartite Entanglement

In quantum chemistry, for a system of basis set $\left\{|n_{p}\rangle \right\}$ with finite cardinality L, each basis $|n_{p}\rangle$ associates with a local Hilbert space $H_{i}$. Then, the total Hilbert space H spanned by the entire basis set is the tensor product of all local Hilbert spaces $H={⊗}_{p=1}^{L}H_{p}$ [24,142,143,144]. For an arbitrary state |Ψ〉∈H(57)$$\begin{array}{cc} |\Psi \rangle & =\sum_{n_{1},\ldots ,n_{L}}C_{n_{1},\ldots ,n_{L}}|n_{1},\ldots ,n_{L}\rangle \\ \; & =\sum_{n_{1},\ldots ,n_{L}}C_{n_{1},\ldots ,n_{L}}|n_{1}\rangle ⊗\ldots ⊗|n_{L}\rangle \end{array}$$

When the system is divided into two parts *A* and *B*, the composite Hilbert space $H\equiv H_{AB}$ of the whole system is given as follows:(58)$$H\equiv H_{\mathrm{AB}}=H_{A}⊗H_{B}$$
where $H_{A}$ and $H_{B}$ are the Hilbert spaces of subsystems *A* and *B*, respectively. Only when there is no entanglement between the systems can the state $|\Psi_{AB}\rangle$ be represented as a tensor product of the states of the subsystems $|\Psi_{A}\rangle$ and $|\Psi_{B}\rangle$, as shown in Figure 8a.(59)$$\;|\Psi \rangle \equiv |\Psi_{AB}\rangle =|\Psi_{A}\rangle ⊗|\Psi_{B}\rangle$$ As a consequence of quantum entanglement among arbitrary subsystems, as shown in Figure 8b, for a generic |Ψ〉∈H, it should be represented as a series of tensor products of basis states of subsystems *A* and *B*,(60)$$\;|\Psi \rangle \equiv |\Psi_{AB}\rangle =\sum_{pq}C_{pq}|\Psi_{A}^{p}\rangle ⊗|\Psi_{B}^{q}\rangle$$

The strategy for measuring bipartite entanglement emerges from the concept of partial measurements [145]. In Section 3.1, we have an introduction of the density matrix for both pure and mixed states; for the sake of brevity, here we only consider the case of pure states. A bipartite reduced density matrix of a pure state can be approached by tracing out (“averaging over”) one of the subsystems:(61)DA=TrB|Ψ〉〈Ψ|DB=TrA|Ψ〉〈Ψ|
thus, DA and DB are the reduced density matrices of each subsystem.

The quantification of bipartite entanglement can be approached through quantum information theory with the von Neumann entropy [141], which quantifies the degree of mixedness of a quantum state, measuring the departure of a given density operator from a pure state (for which S=0, representing complete knowledge of the system) [80]. In the specific case of pure states in bipartite quantum systems, it quantifies the entanglement of the quantum system as(62)S(D)=−Tr(DlnD)
where D is the density matrix. As the quantum counterpart of Shannon entropy we introduced in Equation (Equation 1), the von Neumann entropy can be expressed as the Shannon entropy of the eigenvalues $\lambda_{p}$ for the density matrix:(63)$$S\left(D\right)=-\sum_{p}\lambda_{p}\ln\lambda_{p}$$ To measure the entanglement between the bipartite subsystems, the entanglement entropy $S_{A}$ and $S_{B}$ are defined as follows:(64)SA=−Tr(DAlnDA)(65)SB=−Tr(DBlnDB) For pure states, the relation $S_{A}=S_{B}$ holds, as can be proven via Schmidt decomposition [80].

### 5.2. Orbital Reduced Density Matrix

For a wavefunction expressed in terms of spatial orbitals, each orbital holds four possible occupations, which can be empty |0〉, singly occupied with a spin-up electron |↑〉, singly occupied with a spin-down electron |↑〉, and doubly occupied with an electron pair |↑↓〉, i.e., the possible orbital occupations are(66)$$\begin{array}{c} \{ \end{array}|n_{i}\rangle \begin{array}{c} \} \end{array}=\{|0\rangle ,|↑\rangle ,|↓\rangle ,|↑↓\rangle \}$$ As shown in Figure 9, if the system is divided into subsystem *A* composed of p-orbitals and complement subsystem *B* (orbital bath) composed of the remaining L−p orbitals, the RDM of subsystem *A* is then called the *p*-orbital reduced density matrix (*p*-orbital RDM), which can be defined in terms of the full, *N*-electron RDM or, equivalently, the *N*-electron wavefunction.

The 1-orbital RDM Dpo1 corresponding to the one orbital partition in Figure 9a is expressed in the basis {-, ↑, ↓, ↑↓}, as shown in Table 1 [146,147,148,149], the elements of the matrix can be represented in terms of the spin-dependent 1-electron RDM D¯1 and 2-electron RDM D¯2, where the indices *p* and $\bar{p}$ indicate spin-up and spin-down electrons of *p*-th orbital respectively.

The two orbital partition is shown in Figure 9b, elements of the 2-orbital RDM Dpqo2 are summarized in Table 2 [146,147,148,149]. Note that Dpqo2 requires only some diagonal elements of the 3- and 4-electron reduced density matrices, as well as a few off-diagonal elements of the 1-, 2-, and 3-electron reduced density matrices.

We should note that the 1-orbital RDM Dpo1 can be further simplified for seniority-zero state. Since it excludes singly occupied orbitals and we have the relations D¯pp1=D¯p¯p¯1, the corresponding basis {-, ↑↓} holds a 2 cardinality, thus the 1-orbital RDM of a seniority-zero state is simplified as a 2×2 matrix:(67)Dpo1=1−Dpp100Dpp1 Following the same simplification method, the 2-orbital RDM Dpqo2 of a seniority-zero state expressed in the basis {−, ↑↓, ↑↓, ↑↓↑↓} is reduced to a 4×4 matrix:(68)Dpqo2=1−Dpp1−Dqq1+Dpq¯pq¯20000Dpp1−Dpq¯pq¯2Dqq¯pp¯200Dpp¯qq¯2Dqq1−Dqp¯qp¯20000Dpq¯pq¯2

### 5.3. Orbital Entanglement

With the preliminary knowledge of quantum information theory and orbital reduced density matrix, we can define the single-orbital entropy and mutual information from 1- and 2-orbital RDM. The first quantity, single–orbital entropy $s{\left(1\right)}_{p}$, measures the entanglement between a given orbital *p* and the complementary orbital bath, using the eigenvalues of the one–orbital reduced density matrix as information carriers. The single–orbital entropy $s{\left(1\right)}_{p}$ is defined as follows:(69)s(1)p=−Tr(Dpo1lnDpo1)=−∑αMλα,plnλα,p
where $\lambda_{\alpha ,p}$ and *M* are the eigenvalues and dimension of the *p*-th one-orbital reduced density matrix respectively, with M=2 for seniority-zero state and M=4 for other states. The total quantum information encoded in the system is given by the sum of single-orbital entropy:(70)$$\;I_{tot}=\sum_{p}^{L}s{\left(1\right)}_{p}$$ Given two states described by the one-orbital reduced density matrices Dpo1 and Dqo1, one can define the relative orbital entropy by the KL divergence:(71)s(1)(p||q)=DKL(Dpo1||Dqo1)=Tr(Dpo1lnDpo1Dqo1) It measures how much the entanglement of orbital *p* deviates from that of orbital *q*.

If the system is divided into two orbitals (p,q) and the remaining L−2 orbital bath as shown in Figure 9b, the entanglement between them is quantified by the two-orbital entropy $s{\left(2\right)}_{\left(p,q\right)}$ defined as(72)s(2)(p,q)=−Tr(Dpqo2lnDpqo2)=−∑αMλα,pqlnλα,pq
where $\lambda_{\alpha ,pq}$ and *M* are the eigenvalues and dimension of the two-orbital reduced density matrix respectively, with M=4 for seniority-zero state and M=16 for other states. The conditional orbital entropy can be written by the KL divergence as follows: (73)s(2)(p|q)=−DKL(Dpqo2||Ipo1⊗Dqo1)=−Tr(Dpqo2lnDpqo2Ipo1⊗Dqo1)=s(2)(p,q)−s(1)q where Ipo1 is the identity matrix with the same dimensions as Dpo1. The total amount of entanglement between two orbitals *p* and *q* can be measured by the orbital pair mutual information, written by the KL divergence as (74)I(p;q)=DKL(Dpqo2||Dpo1⊗Dqo1)=Tr(Dpqo2lnDpqo2Dpo1⊗Dqo1)=s(1)p+s(1)q−s(2)(p,q) As an application of the information theory framework introduced in Section 2 where the orbital reduced density matrix serves as the information carrier, those quantum information theory quantities naturally inherit key properties we introduced in Section 2.3, including but not limited to the chain rule and subadditivity.

#### Examples and Illustrations

The concept of orbital entanglement serves as a complementary tool for interpreting electronic structures, proving particularly valuable in strongly correlated systems. In this subsection, we will present several representative application examples.

Electron correlation effects are conventionally categorized as dynamic and nondynamic (also termed static), where in this classification static is used synonymously with nondynamic [156]. The dynamic correlation effect, though large, arise from relatively small contributions from a large number of configurations. In contrast, the nondynamic/static effect involving large contributions arising from a few configurations, which collectively address orbital (near-) degeneracy. In contexts demanding more nuanced differentiation, static electron correlation (strict degeneracy) and nondynamic correlation (near-degeneracy) are considered separate effects in our formalism [148]. As presented in Refs. [47,148], the strength of single-orbital entropy $s{\left(1\right)}_{p}$ to measure orbital entanglement and the orbital-pair mutual information $I_{\left(p;q\right)}$ can be associated with different types of electron correlation effects. In Table 3, we map the strength of orbital interactions onto a certain type of correlation effects. Computational investigations demonstrate that orbitals with nondynamic/static electron correlation effects signify substantial multireference character in the system. In contrast, weakly entangled orbitals predominantly exhibit dynamic correlation effects; systems where all orbitals are weakly entangled can usually be adequately treated by single-reference approaches.

Quantitative visualization of $s{\left(1\right)}_{p}$ and $I_{\left(p;q\right)}$ enables a more intuitive analysis. As shown in Figure 10, the strength of the orbital-pair mutual information classified in Table 3 is color-coded,

Blue lines: Nondynamic correlated orbital pairs.Red lines: Static correlated orbitals.Green: Dynamic correlated orbitals.

An important issue for the density matrix renormalization group (DMRG) method [157,158,159] is the order of orbitals in the one-dimensional matrix product state (MPS) wavefunction ansatz; an optimal ordering of orbitals corresponding to maximum entanglement will produce the most efficient results [21,160,161]. Since strong (nondynamic/static) electron correlation is essential for proper molecular dissociation into fragments, orbital entanglement provides both a fundamental framework for understanding bond formation/breaking processes [150] and a practical tool for analyzing chemical reactivity.

For many strongly correlated calculation methods such as complete active space self-consistent field (CASSCF) [41,162], the selection of the complete active space (CAS space) is a crucial prerequisite step to keep computational costs within feasible limits. As shown in Figure 11, for the CAS methodology, all the molecular orbitals are classified into three spaces:Inactive space: Always doubly occupied.Active space: All the possible configurations are allowed.Virtual space: Always empty.

Specifically, the active space, usually denoted as CAS(n,m) where *n* and *m* are the number of electrons and orbitals respectively, should encompass orbitals and electrons essential for capturing strong electron correlation effects. Orbital entanglement serve as powerful tools for identifying critical orbitals in active space [163,164]. Comparing the entanglement diagrams as shown in Figure 10, we can evaluate the quality and convergence behavior of those active-space calculations.

## 6. Summary and Outlook

This review presents a unified perspective on information theory and its applications in quantum chemistry, integrating both classical and quantum frameworks. Beginning with fundamental concepts like Shannon entropy and its related concepts such as joint entropy, relative entropy, conditional entropy, and mutual information, we demonstrate how information theory can be applied to molecular systems through information carriers such as the electron density and orbital reduced density matrix. The discussion bridges classical concepts with their quantum counterparts, including classical Shannon entropy and quantum correlations to the quantum von Neumann entropy and entanglement. By tracing the historical development from early works by Shannon’s foundational contributions, we highlight how information theory has evolved into a versatile framework with broad applications in quantum chemistry, particularly for analyzing electronic structure and quantum phenomena in chemical systems.

This article only scratches the surface of the vast scope for existing and future applications of information theory in quantum chemistry. Notably, for the application of information theory in quantum chemistry, Shannon’s framework is not the only reasonable formula; the Rényi entropy, the Fisher information, and other *f*- and Bregman divergences can also be used as a measure of correlation and entanglement. Importantly, these concepts extend far beyond simple pairwise interactions: from single- and pair-electron densities to many-body electron distributions, and from bipartite systems to complex multipartite quantum entanglement. The physical manifestations of information carriers are likewise diverse, ranging from atoms in molecules (AIM) to localized molecular orbitals. By creatively combining the above extended concepts as well as other potential extensions, we can even derive additional novel concepts and tools for advancing our understanding of quantum chemistry.

Furthermore, other domains such as machine learning and quantum computing that represent one of the most active research frontiers in science, also possess profound foundations in information theory. The integration of quantum chemistry with information theory, machine learning, and quantum computing is establishing a transformative new paradigm for quantum chemical research.

## Figures and Tables

**Figure 1 entropy-27-00644-f001:**
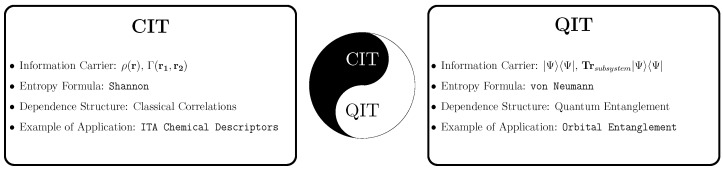
Illustration of some representative concepts of classical information theory (CIT) and quantum information theory (QIT) in quantum chemistry confined to Shannon’s framework.

**Figure 2 entropy-27-00644-f002:**
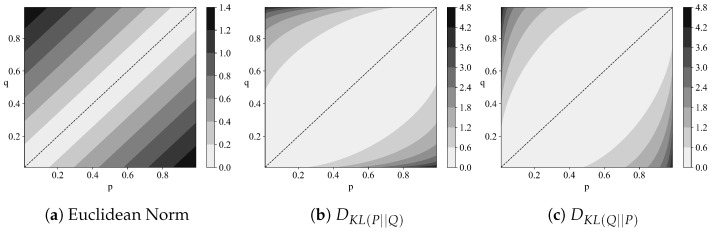
Different measures of distinguishability between two probability distributions P=(p,1−p) and Q=(q,1−q).

**Figure 3 entropy-27-00644-f003:**
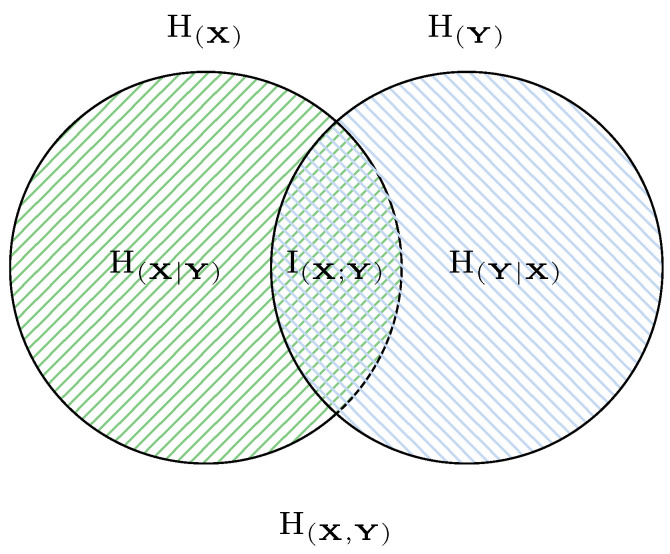
A pseudo Venn diagram illustrating the relationships between Shannon entropy $H_{\left(X\right)}$ or $H_{\left(Y\right)}$ along with consequent concepts: joint entropy $H_{\left(X,Y\right)}$, conditional entropy $H_{\left(X|Y\right)}$ or $H_{\left(Y|X\right)}$, and mutual information $I_{\left(X;Y\right)}$.

**Figure 4 entropy-27-00644-f004:**
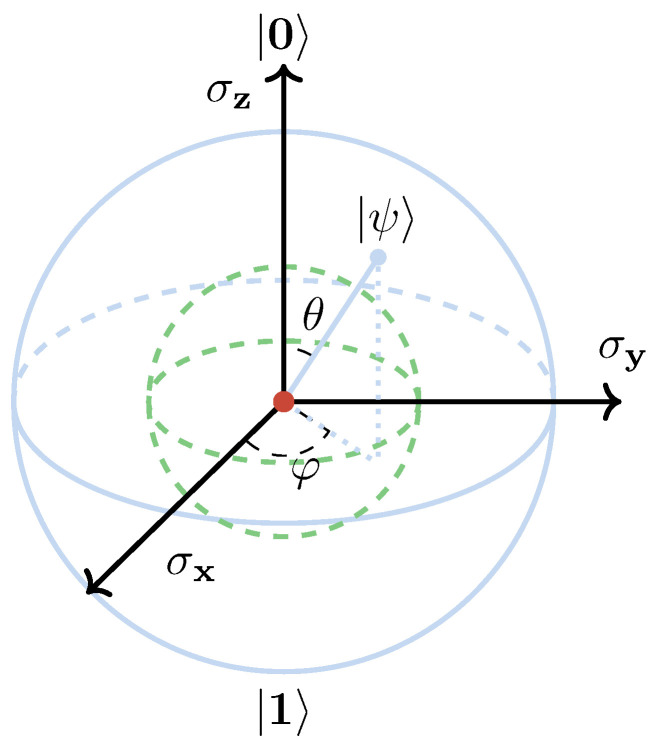
For the Bloch sphere, every point on the surface corresponds to a pure state, every point inside corresponds to a mixed state, and the point located exact center of the Bloch sphere corresponds to a maximally mixed state.

**Figure 5 entropy-27-00644-f005:**
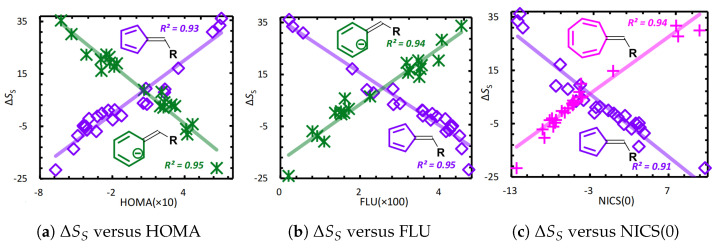
Linear correlation analysis between Shannon entropy aromaticity and several established aromaticity indices. Reproduced with permission from Ref. [121] Copyright 2019, The Author.

**Figure 6 entropy-27-00644-f006:**
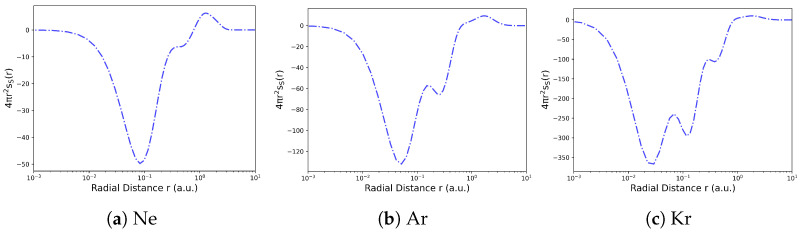
Radial distribution functions of the Shannon entropy densities $4\pi r^{2}s_{s(X)}\left(r\right)$, along the r axis for some noble gas atoms Neon, Argon and Krypton. Reproduced with permission from Ref. [138] Copyright 2025, The Author.

**Figure 7 entropy-27-00644-f007:**
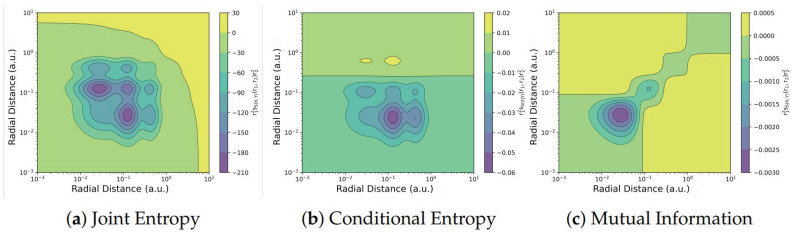
Contour plots of radial distribution function for joint entropy kernel $r_{1}^{2}s_{s(X,Y)}\left(r_{1},r_{2}\right)r_{2}^{2}$, conditional entropy kernel $r_{1}^{2}s_{s(X|Y)}\left(r_{1},r_{2}\right)r_{2}^{2}$ and mutual information kernel $r_{1}^{2}s_{s(X;Y)}\left(r_{1},r_{2}\right)r_{2}^{2}$, for Krypton. Reproduced with permission from Ref. [138] Copyright 2025, The Author.

**Figure 8 entropy-27-00644-f008:**
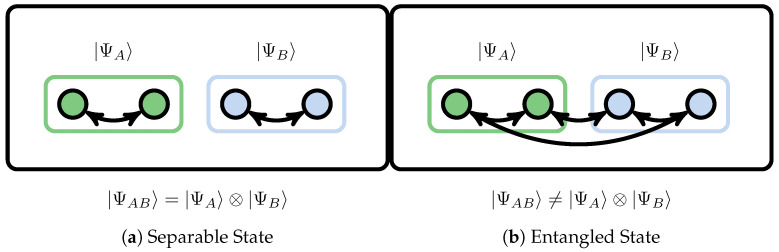
Bipartite of quantum many-body system for the separable state (product state) and entangled state.

**Figure 9 entropy-27-00644-f009:**
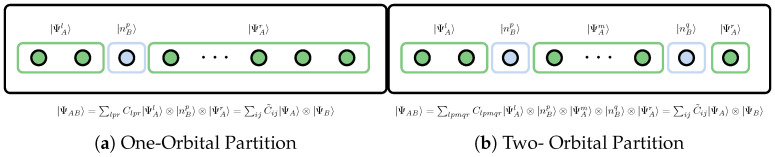
Graph illustration one- and two- orbital bipartite of the system and the corresponding vector state of the composite system AB is displayed.

**Figure 10 entropy-27-00644-f010:**
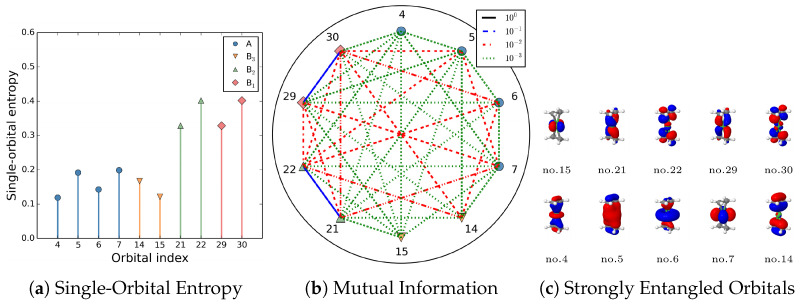
Single-orbital entropy and an alternative definition of orbital mutual information, as proposed in Ref. [148], is given by $I_{\left(p;q\right)}=\frac{1}{2}\left[s{\left(2\right)}_{pq}-s{\left(1\right)}_{p}-s{\left(1\right)}_{q}\right]\left(1-\delta_{pq}\right)$, specifically applied to Ni(C2H4) through DMRG(36,33) calculations. The classified strengths of mutual information in Table 3 are represented using a color code, with the dynamic entanglement effects being disregarded. Reproduced with permission from Ref. [150] Copyright 2015, The Author.

**Figure 11 entropy-27-00644-f011:**
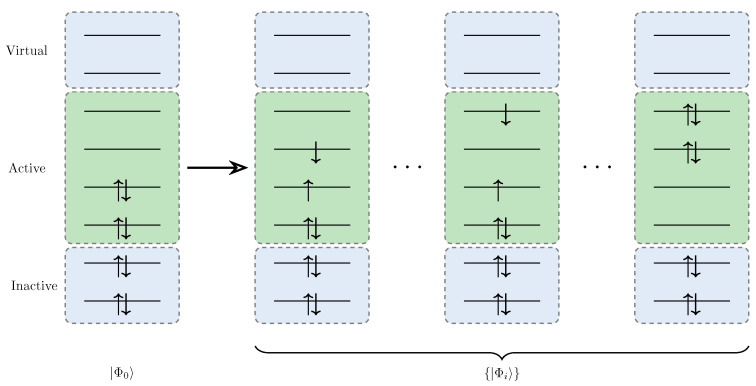
Orbital classification in CAS Methodology: Inactive space remain doubly occupied, active space allow all possible configurations, and virtual space are unoccupied.

**Table 1 entropy-27-00644-t001:** The matrix elements of 1-orbital RDM Dpo1 expressed in terms of 1- and 2-electron RDM.

	-	** ↑ **	** ↓ **	** ↑↓ **
-	1−D¯pp1−D¯p¯p¯1+D¯pp¯pp¯2	0	0	0
↑	0	D¯pp1−D¯pp¯pp¯2	0	0
↓	0	0	D¯p¯p¯1−D¯pp¯pp¯2	0
↑↓	0	0	0	D¯pp¯pp¯2

**Table 2 entropy-27-00644-t002:** The matrix elements of 2-orbital RDM Dpqo2 expressed in terms of 1-, 2-, 3-, and 4-electron RDM.

	-	↑	↑	↓	↓	↑ ↑	↓ ↓	↑↓	↑ ↓	↓ ↑	↑↓	↑↑↓	↑↓↑	↓↑↓	↑↓↓	↑↓↑↓
-	1,1	0	0	0	0	0	0	0	0	0	0	0	0	0	0	0
↑	0	2,2	2,3	0	0	0	0	0	0	0	0	0	0	0	0	0
↑	0	3,2	3,3	0	0	0	0	0	0	0	0	0	0	0	0	0
↓	0	0	0	4,4	4,5	0	0	0	0	0	0	0	0	0	0	0
↓	0	0	0	5,4	5,5	0	0	0	0	0	0	0	0	0	0	0
↑ ↑	0	0	0	0	0	6,6	0	0	0	0	0	0	0	0	0	0
↓ ↓	0	0	0	0	0	0	7,7	0	0	0	0	0	0	0	0	0
↑ ↓	0	0	0	0	0	0	0	8,8	8,9	8,10	8,11	0	0	0	0	0
↑ ↓	0	0	0	0	0	0	0	9,8	9,9	9,10	9,11	0	0	0	0	0
↓ ↑	0	0	0	0	0	0	0	10,8	10,9	10,10	10,11	0	0	0	0	0
↑ ↓	0	0	0	0	0	0	0	11,8	11,9	11,10	11,11	0	0	0	0	0
↑ ↑ ↓	0	0	0	0	0	0	0	0	0	0	0	12,12	12,13	0	0	0
↑ ↓ ↑	0	0	0	0	0	0	0	0	0	0	0	13,12	13,13	0	0	0
↓ ↑ ↓	0	0	0	0	0	0	0	0	0	0	0	0	0	14,14	14,15	0
↑ ↓ ↓	0	0	0	0	0	0	0	0	0	0	0	0	0	15,14	15,15	0
↑ ↓ ↑ ↓	0	0	0	0	0	0	0	0	0	0	0	0	0	0	0	16,16

(1,1)=1−Dpp1−Dp¯p¯1−Dqq1−Dq¯q¯1+Dpp¯pp¯2+Dqq¯qq¯2+Dpqpq2+Dpq¯pq¯2+Dp¯qp¯q2+Dp¯q¯p¯q¯2−Dpqq¯pqq¯3−Dp¯qq¯p¯qq¯3−Dpp¯qpp¯q3−Dpp¯q¯pp¯q¯3+Dpp¯qq¯pp¯qq¯4(2,2)=Dqq1−Dpqpq2−Dp¯qp¯q2−Dqq¯qq¯2+Dpq¯qpq¯q3+Dpp¯qpp¯q3+Dp¯qq¯p¯qq¯3−Dpp¯qq¯pp¯qq¯4(2,3)=(3,2)†=Dpq1−Dpp¯qp¯2−Dpq¯qq¯2+Dpp¯q¯qp¯q¯3(3,3)=Dpp1−Dpp¯pp¯2−Dpqpq2−Dpq¯pq¯2+Dpqq¯pqq¯3+Dpp¯qpp¯q3+Dpp¯q¯pp¯q¯3−Dpp¯qq¯pp¯qq¯4(4,4)=Dq¯q¯1−Dpq¯pq¯2−Dp¯q¯p¯q¯2−Dqq¯qq¯2+Dpp¯q¯pp¯q¯3+Dpqq¯pqq¯3+Dp¯qq¯p¯qq¯3−Dpp¯qq¯pp¯qq¯4(4,5)=(5,4)†=Dp¯q¯1−Dpp¯pq¯2−Dqp¯qq¯2+Dpqp¯pqq¯3(5,5)=Dp¯p¯1−Dp¯qp¯q2−Dp¯q¯p¯q¯2−Dpp¯pp¯2+Dp¯qq¯p¯qq¯3+Dpp¯qpp¯q3+Dpp¯q¯pp¯q¯3−Dpp¯qq¯pp¯qq¯4(6,6)=Dpqpq2−Dpp¯qpp¯q3−Dpqq¯pqq¯3+Dpp¯qq¯pp¯qq¯4(7,7)=Dp¯q¯p¯q¯2−Dpp¯q¯pp¯q¯3−Dp¯qq¯p¯qq¯3+Dpp¯qq¯pp¯qq¯4(8,8)=Dqq¯qq¯2−Dpqq¯pqq¯3−Dpq¯qpq¯q3+Dpp¯qq¯pp¯qq¯4(8,9)=(9,8)†=Dpq¯qq¯2−Dpq¯p¯qq¯p¯3(8,10)=(10,8)†=−Dqp¯qq¯2+Dpqp¯pqq¯3(8,11)=(11,8)†=Dpp¯qq¯2(9,9)=Dpq¯pq¯2−Dpp¯q¯pp¯q¯3−Dpqq¯pqq¯3+Dpp¯qq¯pp¯qq¯4(9,10)=(10,9)†=−Dqp¯pq¯2(9,11)=(11,9)†=Dpp¯pq¯2−Dpqp¯pqq¯3(10,10)=Dp¯qp¯q2−Dpp¯qpp¯q3−Dp¯qq¯p¯qq¯3+Dpp¯qq¯pp¯qq¯4(10,11)=(11,10)†=−Dpp¯qp¯2+Dpp¯p¯qp¯q¯3(11,11)=Dpp¯pp¯2−Dpp¯qpp¯q3−Dpp¯q¯pp¯q¯3+Dpp¯qq¯pp¯qq¯4(12,12)=Dpqq¯pqq¯3−Dpp¯qq¯pp¯qq¯4(12,13)=(13,12)†=−Dpqq¯pqq¯3(13,13)=Dpp¯qpp¯q3−Dpp¯qq¯pp¯qq¯4(14,14)=Dp¯qq¯p¯qq¯3−Dpp¯qq¯pp¯qq¯4(14,15)=(15,14)†=−Dp¯pq¯p¯qq¯3(15,15)=Dpp¯q¯pp¯q¯3−Dpp¯qq¯pp¯qq¯4(16,16)=Dpp¯qq¯pp¯qq¯4.

**Table 3 entropy-27-00644-t003:** Relation between the strength of orbital entanglement and electron correlation effects.

Correlation Effects	Intensity	$s{\left(1\right)}_{p}$	$I_{\left(p;q\right)}$ ^a^
Nondynamic	Strong	>0.5	≈10^−1^
Static	Medium	0.5–0.1	≈10^−1^
Dynamic	Weak	<0.1	≈10^−1^

^a^ An alternative definition of orbital mutual information from Ref. [148] is employed: $I_{\left(p;q\right)}=\frac{1}{2}\left[s{\left(2\right)}_{pq}-s{\left(1\right)}_{p}-s{\left(1\right)}_{q}\right]\left(1-\delta_{pq}\right)$. $\delta_{pq}$ is the Kronecker delta to ensure $I_{\left(p;q\right)}=0$ when p=q.

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
