# Peer review of "Information Theory Meets Quantum Chemistry: A Review and Perspective"

_entropy, 2025, doi:10.3390/e27060644_

Round 1
Reviewer 1 Report
Comments and Suggestions for Authors
This review article offers a clear and accessible introduction to the intersection of information theory and quantum chemistry, aimed at readers seeking foundational knowledge in both fields. The authors begin with a concise presentation of classical information theory based on Shannon entropy and its key extensions, including joint, conditional, and mutual information.
Building on this classical framework, the article transitions into the quantum domain by introducing the concept of reduced density matrices (RDMs)—a critical tool for handling complex many-body quantum states. The manuscript carefully delineates the roles of classical and quantum information measures, showing how classical quantities can be derived from position-representation RDMs (e.g., electron density and pair density), while quantum counterparts like von Neumann entropy serve to characterize entanglement in many-body systems.
By framing both classical and quantum information metrics in the context of quantum chemistry, the paper provides a pedagogically valuable bridge between theoretical physics and computational chemistry. The exposition is methodical and well-structured, offering a roadmap for students and researchers new to this interdisciplinary area. Although the paper is a review rather than a research article, it makes a useful contribution by integrating concepts across disciplines in a coherent and elementary format.
I enjoyed reading this manuscript very much, and filled some gaps in my knowledge, which I believe will be the same for many readers. While it can be accepted for publication as it is, I have two suggestions for improving it.
1- I find Sham’s work on linking entanglement to phase transitions through DFT very interesting. Due to its relevance, I think it would be good to introduce this work as well ( https://doi.org/10.1103/PhysRevA.74.052335 )
2- As a quantum information scientist and also working in DFT and using Quantum Espresso (QE), I would like to see if/how QE or similar other software can be used for quantum information with DFT. So, a discussion on opportunities and limitations of the existing software, as well as suggestions for improvements of these software would be beneficial, I think.
I checked the similarity report in detail due to high rate (24%), but it is due to some standard expressions, and I confirm that this manuscript is free of any such issues.
The English of the manuscript is fine, I did not detect any issue. It was very smooth to read the entire manuscript -even with a critical point of view, as a referee.
The literature is covered very well -though I suggested one more work above.
Figures are very beneficial for non-quantum information scientists to grasp the basic concepts.
Reviewer 2 Report
Comments and Suggestions for Authors
See attached pdf for comments.
